# Hematological Ratios Are Associated with Acute Kidney Injury and Mortality in Patients That Present with Suspected Infection at the Emergency Department

**DOI:** 10.3390/jcm11041017

**Published:** 2022-02-16

**Authors:** Titus A. P. de Hond, Gurbey Ocak, Leonie Groeneweg, Jan Jelrik Oosterheert, Saskia Haitjema, Meriem Khairoun, Karin A. H. Kaasjager

**Affiliations:** 1Department of Internal Medicine and Acute Medicine, University Medical Centre Utrecht, Utrecht University, 3584 CX Utrecht, The Netherlands; l.groeneweg@umcutrecht.nl (L.G.); h.a.h.kaasjager@umcutrecht.nl (K.A.H.K.); 2Department of Internal Medicine, Sint Antonius Hospital, 3435 CM Nieuwegein, The Netherlands; g.ocak@antoniusziekenhuis.nl; 3Department of Internal Medicine and Infectious Diseases, University Medical Centre Utrecht, Utrecht University, 3584 CX Utrecht, The Netherlands; j.j.oosterheert@umcutrecht.nl; 4Central Diagnostic Laboratory, University Medical Centre Utrecht, Utrecht University, 3584 CX Utrecht, The Netherlands; s.haitjema@umcutrecht.nl; 5Department of Nephrology and Hypertension, University Medical Centre Utrecht, Utrecht University, 3508 CX Utrecht, The Netherlands; m.khairoun@umcutrecht.nl

**Keywords:** hematological ratios, emergency department, acute kidney injury, mortality, infection, inflammation

## Abstract

The early recognition of acute kidney injury (AKI) is essential to improve outcomes and prevent complications such as chronic kidney disease, the need for renal-replacement therapy, and an increased length of hospital stay. Increasing evidence shows that inflammation plays an important role in the pathophysiology of AKI and mortality. Several inflammatory hematological ratios can be used to measure systemic inflammation. Therefore, the association between these ratios and outcomes (AKI and mortality) in patients suspected of having an infection at the emergency department was investigated. Data from the SPACE cohort were used. Cox regression was performed to investigate the association between seven hematological ratios and outcomes. A total of 1889 patients were included, of which 160 (8.5%) patients developed AKI and 102 (5.4%) died in <30 days. The Cox proportional-hazards model revealed that the neutrophil-to-lymphocyte ratio (NLR), segmented-neutrophil-to-monocyte ratio (SMR), and neutrophil-lymphocyte-platelet ratio (NLPR) are independently associated with AKI <30 days after emergency-department presentation. Additionally, the NLR, SMR and NLPR were associated with 30-day all-cause mortality. These findings are an important step forward for the early recognition of AKI. The use of these markers might enable emergency-department physicians to recognize and treat AKI in an early phase to potentially prevent complications.

## 1. Introduction

Acute kidney injury (AKI) is a group of syndromes characterized by a rapid decline of renal function [1,2,3]. The incidence of AKI varies depending on the population and the definition used, and has steadily increased over the past few decades [3,4]. AKI is common in critically ill septic patients and is associated with several short- and long-term outcomes, including an increased length of hospital stay, morbidity (e.g., the development of end-stage renal disease with the need for renal-replacement therapy (RRT)), and mortality. Altogether, this results in the impairment of the sustainability of the healthcare system [5,6,7,8].

AKI has a complex pathophysiology and in most cases the development is multifactorial. Severe infections and sepsis in critically ill patients make up an important part of the causes of AKI, and there is growing evidence that even patients with less-severe infections have a higher risk of developing AKI [9,10]. Inflammation may play an important role in the pathogenesis of AKI in patients with severe infections or sepsis, and evidence shows that AKI can even occur in the absence of hypoperfusion [11]. However, the early recognition and treatment of AKI remains difficult. Therefore, it is important to investigate the risk factors for AKI. Currently, the diagnosis of AKI is mainly dependent on the use of serum creatinine (SCr), which can be relatively delayed and is affected by many factors [12].

Indeed, several studies have focused on biomarkers (e.g., KIM-1, NGAL) to detect AKI in the early phase [13]. However, single biomarkers were proven not to be sensitive enough for the detection of AKI and were not routinely available in daily clinical practice. Recent studies described complete blood count to be associated with AKI [13]. Several hematological ratios that can be calculated from complete blood count (CBC) are associated with inflammation [14,15,16]. As local and systemic inflammation play an important role in the initiation and progression of AKI and mortality in patients with infections, hematological ratios could be associated with AKI and mortality in patients with infections. Therefore, the aim of this study was to investigate the association between inflammatory hematological ratios, which were derived from a routine CBC, and the occurrence of AKI and mortality in patients suspected of having an infection at the emergency department.

## 2. Materials and Methods

### 2.1. Study Design

This prospective observational cohort study was performed within the University Medical Center Utrecht (UMCU). The UMCU is a large, tertiary academic teaching hospital located in the Netherlands. Patients described in this study were included between 13 September 2016 and 1 January 2019 and followed until the time of death or censored due to loss of a follow-up, or until one year after their emergency-department presentation.

Clinical data were used from the SPACE cohort (SePsis in acutely ill patients in the emergency room) [17]. This study was reviewed and approved by the Medical Ethical Committee of the UMCU under number 16/594 and registered in the Dutch Trial Register (NTR) under number 6916. Since only pseudonymized data were included, the review board decided that written patient consent was not required. This study was performed in accordance with the Declaration of Helsinki.

### 2.2. Study Population and Data Collection

The SPACE cohort consists of all consecutive patients who meet the following criteria: (1) ≥18 years or older; (2) presentation at the emergency department with a suspected infection defined by the treating physician; (3) registration in the emergency department for the internal-medicine department or one of its subspecialties of oncology, rheumatology, immunology, hematology, nephrology, endocrinology, geriatrics, infectious disease or vascular medicine [17]. For the current analyses, patients receiving RRT, patients without baseline or follow-up SCr and patients that had a repeated emergency-department visit within 30 days were excluded.

Demographics, clinical data recorded at the time of presentation in the emergency department and, if applicable, during hospitalization, as well as data on laboratory markers, treatments, and outcomes were collected for all eligible patients in this cohort. Data on immune status, comorbidities, diagnosis at admission, and diagnosis at discharge were manually extracted from the Electronic Health Records (EHR) by independent researchers and reviewed using a set of predefined and standardized definitions. All available data on SCr within 30 days after presentation were extracted from the EHR. The qSOFA score was calculated by assigning one point each for respiratory rate ≥22 breaths/min, systolic blood pressure ≤100 mm Hg and altered mental status) and was considered positive in the case of a score ≥2 [18]. Comorbidities were quantified using the Charlson Comorbidity Index (CCI) [19].

Additional data were retrieved using the Utrecht Patient Oriented Database (UPOD). The structure and content of UPOD have been described in more detail elsewhere [20]. Hematological markers were retrieved from the UPOD that were determined by automated blood-cell analyses performed with the Abbot Cell-Dynn Sapphire hematology analyzer (Abbott Hematology, Santa Clara, CA, USA). The UPOD setup saves a wide range of hematological parameters, even if they are not specifically requested by the physician [20].

### 2.3. Hematological Ratios

From all patients, blood samples that were obtained during presentation in the emergency department were analyzed to determine the complete blood count as part of the routine clinical care. The hematological ratios were then calculated from the results. The modified delta-neutrophil index (modified DNI) can be determined by subtracting the fraction of segmented neutrophils from the sum of the fractions of total neutrophils and eosinophils. The neutrophil-to-lymphocyte ratio (NLR) (neutrophil count/lymphocyte count) [14], monocyte-to-lymphocyte ratio (MLR) (monocyte count/lymphocyte count) [15], segmented-neutrophil-to-monocyte ratio (SMR) (segmented neutrophil count/monocyte count), platelet-to-lymphocyte ratio (PLR) (platelet count/lymphocyte count) [16], neutrophil-to-lymphocytes-and-platelets ratio (NLPR) ((neutrophil count × 100)/(lymphocyte count × platelet count)) [21] and systemic immune-inflammation (SII) index (platelet count × (neutrophil count/lymphocyte count)) [22] were all calculated from the complete blood count. The modified DNI and SMR were inspired by the previously described delta-neutrophil index (DNI) and segmented-neutrophil-to-mature-monocyte ratio (SeMo), respectively [23,24]. However, since an incomparable method was used to determine leukocyte maturity, other names were chosen to describe these ratios.

### 2.4. Outcomes and Definitions

The primary outcome in this study was the development of AKI within 30 days after presentation at the emergency department. The diagnosis and staging of AKI were based on the Kidney Disease: Improving Global Outcomes (KDIGO) criteria, and were defined as an increase in SCr of 1.5 times the baseline value or an increase in SCr of 26.5 umol/L relative to the value measured at the emergency department within the first 48 h after presentation (stage 1) [2]. Baseline creatinine was defined as the most recent SCr measurement available from 7 days to 12 months prior to presentation at the emergency department. The median period between baseline SCr and presentation at the emergency department was 24 days (interquartile range [IQR] 13–56). The highest SCr within 30 days was used to stratify the KDIGO stages of AKI, with an increase of 2.0 to 2.9 times baseline SCr being classified as stage 2, and an increase of >3.0 times baseline SCr or 352.6 umol/L being classified as stage 3.

The secondary outcome was defined as all-cause mortality within 30 days after emergency-department presentation. Sensitivity analyses for AKI and all-cause mortality within 14 days after emergency-department presentation were included in order to evaluate the robustness of the results.

### 2.5. Statistical Analyses

Continuous variables are presented as medians and interquartile ranges and categorical variables are presented as frequencies and percentages. Non-normally distributed variables were compared using the Mann–Whitney U test. Cox proportional-hazard models were used to calculate hazard ratios (HRs) with 95% confidence intervals (CIs) in order to assess the association between the different hematological ratios and outcomes (AKI and mortality). Hematological ratios were divided into tertiles, with the lowest tertile used as a reference. Hazard ratios were adjusted for potential confounders (based on the literature) including age, sex, Charlson Comorbidity Index, immunocompromised, use of medication (angiotensin-converting-enzyme inhibitors (ACEi) or angiotensin-receptor blockers (ARBs), diuretics, proton-pump inhibitors (PPIs) and non-steroidal anti-inflammatory drugs (NSAIDs)), disease severity (qSOFA) and the provisional diagnosis in the emergency department. Sensitivity analyses were performed using the same Cox proportional-hazard models with different time endpoints for the outcomes. Statistical significance was defined at a *p*-value <0.05. Statistical analyses were performed with the statistical software package SPSS 25.0 for Windows (IBM Corp, Armonk, NY, USA).

## 3. Results

### 3.1. Study Population and Baseline Characteristics

From 3,669 patients, a total of 1,889 patients remained eligible for the final analysis (Figure 1). The baseline characteristics of the included patients are shown in Table 1. The median age was 62 years (IQR 50–70) and 54.5% of the patients were male. Lower-respiratory-tract infection (22.0%) was the most common diagnosis in the emergency department, followed by urinary-tract infection (17.7%) and viral respiratory-tract infection (15.5%). Of all the patients, 821 (43.3%) were immunocompromised and 98 (5.2%) were considered critically ill based on a qSOFA score of ≥2.

### 3.2. Incidence of AKI

Among the 1,889 patients that were analyzed, 160 (8.5%) developed AKI as defined by the KDIGO criteria within the first 30 days after presentation at the emergency department. The median time between ED presentation to the occurrence of AKI was 5 days (IQR 2–14). Within the AKI group, there were 123 (76.9%) patients with AKI stage 1, 20 (12.5%) patients with AKI stage 2, and 17 (10.6%) patients with AKI stage 3.

### 3.3. Hematological Ratios in AKI and Non-AKI Patients

The hematological ratios calculated from the CBC and the distribution between the AKI and non-AKI patients are shown in Table 2. The NLR (median AKI group 8.52, IQR 4.38–17.92 vs. median non-AKI group 6.80, IQR 3.17–12.73), SMR (median AKI group 10.97, IQR 7.03–17.10 vs. median non-AKI group 8.83, IQR 5.62–13.88) and NLPR (median AKI group 4.84, IQR 2.01–10.24 vs. median non-AKI group 3.03, IQR 1.45–6.55) were significantly higher in the AKI group compared to the non-AKI group. Additionally, the distributions of these three ratios differed significantly between the several stages of AKI (Appendix A).

### 3.4. Associations between Hematological Ratios and AKI

Table 3 shows the hazard ratios for several hematological ratios and the occurrence of AKI <30 days after the emergency-department visit. In the univariate Cox regression, there was a significant association between AKI and the highest tertiles of the NLR, SMR, NLPR and SII index. After adjustment for age, sex, comorbidities, baseline renal function, immune status, medication use, disease severity and diagnosis in the emergency department, the highest tertile of the NLR (adjusted HR 1.8, 95% confidence interval [95% CI] 1.2–2.8), middle tertile of the SMR (adjusted HR 1.7; 95% CI 1.1–2.6), highest tertile of the SMR (adjusted HR 2.0; 95% CI 1.3–3.0) and highest tertile of the NLPR (adjusted HR 2.1; 95% CI 1.4–3.2) remained independently associated with the occurrence of AKI <30 days after emergency-department presentation. In a continuous analysis, NLR and SMR were significant as well (adjusted HR 1.002; 95% CI 1.001–1.002 and adjusted HR 1.005; 95% CI 1.003–1.008, respectively).

### 3.5. Association between Hematological Ratios and Mortality

Of the 1889 patients, 102 died within 30 days (5.4%). The analysis of the Cox proportional-hazard models demonstrated the association between several hematological ratios and 30-day mortality (Table 4). After adjustment for confounders, the highest tertile of the NLR (Adjusted HR 1.7; 95% CI 1.0–2.9), highest tertile of the SMR (adjusted HR 1.8; 95% CI 1.1–3.1), highest tertile of the PLR (adjusted HR 1.7; 95% CI 1.1–2.8), middle and highest tertiles of the NLPR (adjusted HR 2.5; 95% CI 1.4–4.4 and adjusted HR 2.1; 95% CI 1.2–3.7), and highest tertile of the SII index (adjusted HR 1.8; 95% CI 1.1–2.8) were shown to be independently associated with 30-day mortality as compared with the lowest tertiles. In a continuous analysis, NLPR was significantly associated with 30-day mortality (adjusted HR 1.008; 95% CI 1.003–1.013).

### 3.6. Sensitivity Analysis

As a sensitivity analysis, we used the same models to examine the associations of the hematological ratios with the occurrence of AKI <14 days after emergency-department presentation. The highest tertile of the NLR (adjusted HR 2.1; 95% CI 1.3–3.5), highest tertile of the SMR (adjusted HR 2.1; 95% CI 1.3–3.4) and highest tertile of the NLPR (adjusted HR 2.4; 95% CI 1.5–3.8) were associated with an increased risk of developing AKI as compared with the lowest tertiles (Appendix A, Appendix A).

The sensitivity analysis showed that he highest tertile of the SMR (adjusted HR 2.3, 95% CI 1.0–5.4), and middle and highest tertiles of the NLPR (adjusted HR 2.6, 95% CI 1.1–6.3 and adjusted HR 2.6, 95% CI 1.1–6.3) were associated with 14-day mortality as compared with the lowest tertiles. The middle tertile of modified DNI (adjusted HR 0.3; 95% CI 0.1–0.7) was associated with a decreased risk of mortality.

## 4. Discussion

In the current study, the inflammatory hematological ratios NLR, SMR and NLPR were associated with an increased risk of developing AKI <30 days after admission to the emergency department. In addition, increased values of the NLR, SMR, PLR, NLPR and the SII index were independently associated with a higher risk of 30-day mortality.

Several prior studies that investigated the association between inflammatory hematological ratios and AKI showed similar results. A study of a single emergency-room measurement of the NLR reported a cut-off value of 5.5 to be associated with AKI (odds ratio 6.4; 95% CI 2.7–16) [25]. In septic patients, two retrospective studies also demonstrated that NLR was an important risk factor for the occurrence of AKI [26,27]. Furthermore, there are several studies that showed an association of the occurrence of AKI with elevated postoperative or post-coronary angiography NLR levels, implicating the important role of inflammation in the development of AKI [28,29,30]. Due to a lack of previous studies on the association between the SMR, NLPR, and AKI, it is not possible to compare our results with previous data. In the present study, there was no association between the modified DNI, MLR, PLR and SII index and the occurrence of AKI. Although there are some previous studies that suggested that these ratios are associated with AKI, they were performed in other settings (e.g., postoperative or ICU) or with other types of ratio measurements (e.g., DNI requires measuring myeloperoxidase-reactive cells and a nuclear lobularity assay) [28,31,32,33,34,35].

In the context of mortality, multiple studies have shown that high NLR values are associated with mortality [36,37]. A retrospective study of critically ill AKI patients also reported higher NLR levels to be associated with all-cause mortality (HR 1.83; 95% CI 1.66–2.02) [38]. There are a few studies that investigated the association between some sort of neutrophil-monocyte ratio and mortality, which found contradictive results. One study investigated neutrophil-to-monocyte ratio and found a high value (≥17.75) to be associated with mortality in COVID-19 patients [39]. Another study investigated the ratio between segmented neutrophils and mature monocytes (SeMo). This study reported the opposite: a SeMo ratio of <16 in critically ill septic patients was associated with a higher risk of 28-day mortality [23]. A likely explanation for the difference in findings between the study of Fang et al. and our study is the use of a different analyzer with incomparable measuring methods. Additionally, the current study had a different study population. In the study of Fang et al., only patients with severe sepsis or septic shock who were already admitted to the ICU were included, compared to only 98 (5.2%) patients in our population that were considered critically ill based on the qSOFA score. For NLPR, there are several studies that showed the association between higher perioperative levels and postoperative AKI and mortality [21,40,41]. Furthermore, another study indicated that an NLPR of ≥15.48 was associated with 28-day mortality in septic patients, which is concordant with the association of higher NLPR values with AKI and mortality in the present study [42]. In line with the current findings, several studies found an association between high PLR levels, high SII-index values and mortality [33,34,43,44].

There are several potential pathophysiological reasons why these inflammatory hematological ratios are associated with AKI. The pathophysiology of AKI in severe infections and sepsis is complex and multifactorial [9]. Septic AKI is often attributed to an ischemia–reperfusion injury resulting from hypoperfusion and shock. However, studies have found that AKI in patients with infections and sepsis can even occur in the absence of hypoperfusion [10,45]. Moreover, it was shown that an inflammatory response was even present in patients with non-severe infections and AKI [10,46]. The role of inflammation in AKI is still not completely understood, but it is likely that pro-inflammatory changes in the endothelial and epithelial cells of the kidney play an important role [47,48]. This is caused by a complex inflammatory process that is triggered by both the innate and adaptive immune system. In the initial phase, renal epithelial cells increase the expression of damage-associated pattern (DAMP) molecules and Toll-like receptors (TLRs). This induces a recruitment of innate immune cells, including neutrophils and macrophages [49,50]. At the same time, macrophages and neutrophils are activated by natural killer T cells that transfer from the vascular system to the renal tissue [51]. Neutrophils are mainly involved during the first 24 h by endothelium adherence and the release of cytokines, reactive oxygen species, and proteases [52,53]. This is followed by a reaction of the adaptive immune system in which lymphocytes cause cellular damage in combination with the release of pro-inflammatory cytokines [52,53].

The present study has several strengths. Our study included a large and well-defined cohort of patients. The SPACE cohort represents a similar population to the population that is generally seen in the emergency department, which makes the results of this study clinically relevant. Additionally, to our knowledge, this is the first study that investigated which hematological ratios are associated with AKI in patients suspected of having an infection at the emergency department. There are two studies that investigated the association between hematological ratios and the occurrence of AKI at the emergency department [25,31]. However, these studies did not specifically investigate patients that were suspected of having an infection and both studies only investigated a single ratio. Furthermore, sensitivity analyses with different time endpoints for the same outcomes showed similar results, which contributes to the robustness of the main results. Nonetheless, this study also has several limitations. Firstly, although some of the data in the SPACE cohort were collected prospectively, data on SCr levels were collected retrospectively and therefore the occurrence of the primary endpoint of AKI might possibly be underestimated, as only patients with available follow-up SCr data could be included in the analyses. We had no data available on urinary output, limiting our definition of AKI. Additionally, it is known in mice that sepsis reduces muscle perfusion. Consequently, the production of creatinine falls, which limits the usability of SCr for early detection of AKI in septic patients [54]. On top of that, AKI could be underdiagnosed due to dilutional effects of fluid-resuscitation therapy. Secondly, it is important to point out that this study was conducted in a tertiary-care institution and that a considerable amount of the patients was immunocompromised. This might make results less generalizable to other settings, since these patients might have a less-reactive immune response. Nonetheless, we corrected for immune status in our analyses and therefore expect this to have minor influence on our final results. Lastly, we did not have data on cause of death within the SPACE cohort. Therefore, although this would have been interesting, we were unable to investigate to what extent AKI contributed to mortality.

The use of inflammatory hematological ratios as additions to the current practice could be useful for the initiation of early treatment such as the discontinuation of nephrotoxic agents, optimization of volume status and perfusion pressure, and monitoring serum creatinine and urine output [55]. This could prevent or limit the extent of short- and long-term outcomes such as the development of chronic kidney disease or end-stage renal disease with the need for costly and life-impacting interventions such as RRT [6,7]. Nonetheless, the diagnosis of AKI mostly depends on the use of traditional indicators such as serum creatinine, which is fairly delayed and is often increasing when AKI is already present [56]. Blood-count parameters such as NLR, SMR and NLPR are easily calculated since blood counts are measured in most patients at the emergency department when there is a suspicion of infection. Therefore, these ratios are easy applicable in the emergency-department setting with low costs, even in patients with less-severe infections who might still be at risk of developing AKI.

In conclusion, the inflammatory hematological ratios NLR, SMR and NLPR are associated with an increased risk of AKI and mortality in patients with suspected infections at the emergency department. This is an important step forward for the early recognition of AKI and might enable physicians to initiate treatment in an early phase. In the future, additional prospective studies that investigate the implementation of these ratios in clinical models are needed.

## Figures and Tables

**Figure 1 jcm-11-01017-f001:**
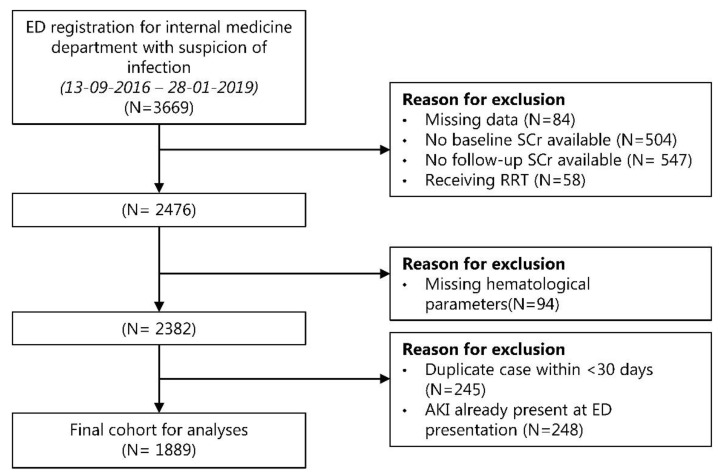
Flowchart of study attrition. Abbreviations: AKI, acute kidney injury; ED, emergency department; RRT, renal replacement therapy; SCr, serum creatinine.

**Table 1 jcm-11-01017-t001:** Baseline characteristics of study population at emergency department by AKI status.

	Total (N = 1889)	No AKI (N = 1729)	AKI (N = 160)
**Demographics**			
Age (years), median (IQR)	62 (50–70)	62 (49–70)	64 (54–72)
Sex (male), *n* (%)	1029 (54.5)	938 (54.3)	91 (56.9)
**Comorbidities**			
Charlson Comorbidity Index, median (IQR)	5.0 (3.0–7.0)	5.0 (3.0–7.0)	5.0 (4.0–7.8)
Hypertension, *n* (%)	618 (32.7)	551 (31.9)	67 (41.9)
Diabetes Mellitus, *n* (%)	367 (19.4)	324 (18.7)	43 (26.9)
Sever liver disease, *n* (%)	20 (1.1)	18 (1.0)	2 (1.3)
Congestive heart failure, *n* (%)	105 (5.1)	88 (5.6)	17 (10.6)
Myocardial infarction, *n* (%)	174 (9.2)	159 (9.2)	15 (9.4)
Peripheral vascular disease, *n* (%)	144 (7.6)	125 (7.2)	19 (11.9)
Cerebrovascular disease, *n* (%)	203 (10.7)	178 (10.3)	25 (15.6)
Kidney Transplant, *n* (%)	254 (13.9)	226 (13.1)	28 (17.5)
Immunocompromised, *n* (%)	821 (43.5)	748 (43.3)	73 (45.6)
Chronic kidney disease, *n* (%)			
Stage 1	6 (0.3)	4 (0.2)	2 (1.3)
Stage 2	81 (4.3)	78 (4.5)	3 (1.9)
Stage 3	270 (14.3)	240 (13.9)	30 (18.8)
Stage 4	115 (6.1)	97 (5.6)	18 (11.3)
Stage 5	39 (2.1)	28 (1.6)	11 (6.9)
**Medication**			
ACE inhibitor or angiotensin-receptor blockers, *n* (%)	501 (26.5)	448 (25.9)	53 (33.1)
Diuretics, *n* (%)	389 (20.6)	345 (20.0)	44 (27.5)
PPI, *n* (%)	1,003 (53.1)	913 (52.8)	90 (56.3)
NSAID, *n* (%)	76 (4.0)	72 (4.2)	4 (2.5)
**Disease severity in ED**			
qSOFA-score ≥2, *n* (%)	98 (5.2)	86 (5.0)	12 (7.5)
**Provisional diagnosis in the ED**			
Lower-respiratory-tract infection, *n* (%)	415 (22.0)	374 (21.6)	41 (25.6)
Viral respiratory-tract infection, *n* (%)	293 (15.5)	281 (16.3)	12 (7.5)
Urinary-tract infection, *n* (%)	334 (17.7)	299 (17.3)	35 (21.9)
Gastro-intestinal infection, *n* (%)	262 (13.9)	247 (14.3)	15 (9.4)
Skin infection, *n* (%)	131 (6.9)	122 (7.1)	9 (5.6)
Other (infectious) diagnosis, *n* (%)	454 (24.0)	406 (23.5)	48 (30.0)
**Laboratory**			
Baseline serum creatinine (umol/L)	79 (63.0–116.0)	78.0 (63.0–113.0)	99.0 (63.3–161.5)
Baseline eGFR CKD-EPI (ml/min)	80.6 (52.3–98.9)	81.4 (53.9–99.1)	61.9 (33.7–97.0)

Abbreviations: ACE, angiotensin-converting enzyme; AKI, acute kidney injury; CKD-EPI, Chronic Kidney Disease Epidemiology Collaboration; ED, emergency department; eGFR, estimated glomerular-filtration rate; IQR, interquartile range; NSAID, non-steroidal anti-inflammatory drugs; qSOFA, quick Sepsis-Related Organ-Failure Assessment.

**Table 2 jcm-11-01017-t002:** Distribution of hematological ratios in AKI vs. non-AKI patients.

	Total (N = 1889)	No AKI (N = 1729)	AKI (N = 160)	*p*-Value
Modified delta-neutrophil index (IQR)	1.17 (0.41–4.73)	1.18 (0.41–4.47)	1.15 (0.38–7.89)	0.64
Neutrophil-lymphocyte ratio (IQR)	6.92 (3.27–13.04)	6.80 (3.17–12.73)	8.52 (4.38–17.92)	<0.001
Monocyte-lymphocyte ratio (IQR)	0.70 (0.40–1.19)	0.70 (0.40–1.19)	0.70 (0.43–1.31)	0.43
Segmented-neutrophil-monocyte ratio (IQR)	8.99 (5.73–14.11)	8.83 (5.62–13.88)	10.97 (7.03–17.10)	<0.001
Platelet-lymphocyte ratio (IQR)	225.36 (137.40–387.90)	225.06 (136.91–382.96)	243.85 (139.47–439.62)	0.50
Neutrophil-lymphocyte-and-platelets ratio (IQR)	3.15 (1.51–6.82)	3.03 (1.45–6.55)	4.84 (2.01–10.24)	<0.001
Systemic immune-inflammation index (IQR)	1469.48 (609.94–3226.94)	1463.68 (586.76–3177.88)	1612.76 (792.82–3858.33)	0.08

Abbreviations: AKI, acute kidney injury; IQR, interquartile range.

**Table 3 jcm-11-01017-t003:** HRs (95% CI) for AKI <30 days after ED presentation.

	Univariate	Multivariate			
Ratios	Crude HR (95% CI)	Adjusted HR: Model 1 ^a^ (95% CI)	Adjusted HR: Model 2 ^b^ (95% CI)	Adjusted HR: Model 3 ^c^ (95% CI)	Adjusted HR: Model 4 ^d^ (95% CI)
**Modified DNI**					
Tertile 1, ≤0.5931	1.0 (reference)	1.0 (reference)	1.0 (reference)	1.0 (reference)	1.0 (reference)
Tertile 2, 0.5931–2.7088	0.7 (0.5–1.0)	0.7 (0.5–1.0)	0.7 (0.5–1.0)	0.7 (0.5–1.0)	0.7 (0.5–1.1)
Tertile 3, >2.7088	1.0 (0.7–1.4)	1.0 (0.7–1.4)	1.0 (0.7–1.4)	1.0 (0.7–1.4)	1.0 (0.7–1.4)
**NLR**					
Tertile 1, ≤4.2805	1.0 (reference)	1.0 (reference)	1.0 (reference)	1.0 (reference)	1.0 (reference)
Tertile 2, 4.2805–10.2276	1.4 (1.0–2.2)	1.4 (1.0–2.1)	1.4 (1.0–2.1)	1.3 (0.9–2.1)	1.3 (0.9–2.0)
Tertile 3, >10.2276	**2.1 (1.4–3.1)**	**2.0 (1.4–3.0)**	**2.0 (1.3–2.9)**	**1.9 (1.3–2.9)**	**1.8 (1.2–2.8)**
**MLR**					
Tertile 1, ≤0.5057	1.0 (reference)	1.0 (reference)	1.0 (reference)	1.0 (reference)	1.0 (reference)
Tertile 2, 0.5057–0.9830	1.1 (0.8–1.7)	1.1 (0.7–1.6)	1.1 (0.7–1.6)	1.0 (0.7–1.6)	1.0 (0.7–1.5)
Tertile 3, >0.9830	1.2 (0.9–1.8)	1.2 (0.8–1.7)	1.2 (0.8–1.7)	1.2 (0.8–1.7)	1.1 (0.8–1.7)
**SMR**					
Tertile 1, ≤6.7500	1.0 (reference)	1.0 (reference)	1.0 (reference)	1.0 (reference)	1.0 (reference)
Tertile 2, 6.7500–11.9633	**1.8 (1.2–2.8)**	**1.8 (1.2–2.7)**	**1.8 (1.2–2.7)**	**1.7 (1.1–2.7)**	**1.7 (1.1–2.6)**
Tertile 3, >11.9633	**2.2 (1.4–3.3)**	**2.2 (1.4–3.3)**	**2.1 (1.4–3.2)**	**2.0 (1.3–3.1)**	**2.0 (1.3–3.0)**
**PLR**					
Tertile 1, ≤161.6468	1.0 (reference)	1.0 (reference)	1.0 (reference)	1.0 (reference)	1.0 (reference)
Tertile 2, 161.6468–314.6356	0.9 (0.6–1.3)	0.9 (0.6–1.3)	0.8 (0.6–1.3)	0.8 (0.6–1.2)	0.8 (0.6–1.3)
Tertile 3, 314.6356	1.1 (0.8–1.6)	1.1 (0.7–1.6)	1.1 (0.7–1.5)	1.1 (0.7–1.5)	1.0 (0.7–1.5)
**NLPR**					
Tertile 1, ≤1.9151	1.0 (reference)	1.0 (reference)	1.0 (reference)	1.0 (reference)	1.0 (reference)
Tertile 2, 1.9151–5.0605	1.4 (1.0–2.2)	1.4 (0.9–2.2)	1.4 (0.9–2.2)	1.4 (0.9–2.1)	1.3 (0.9–2.0)
Tertile 3, >5.0605	**2.4 (1.6–3.5)**	**2.3 (1.5–3.4)**	**2.2 (1.5–3.2)**	**2.2 (1.4–3.3)**	**2.1 (1.4–3.2)**
**SII index**					
Tertile 1, ≤869.43	1.0 (reference)	1.0 (reference)	1.0 (reference)	1.0 (reference)	1.0 (reference)
Tertile 2, 869.43–2414.46	1.4 (1.0–2.0)	1.3 (0.9–2.0)	1.3 (0.9–2.0)	1.3 (0.9–1.9)	1.3 (0.8–1.9)
Tertile 3, >2414.46	**1.6 (1.1–2.4)**	**1.5 (1.0–2.3)**	**1.5 (1.0–2.3)**	**1.5 (1.0–2.2)**	**1.4 (0.9–2.1)**

Abbreviations: HR, hazard ratio; AKI, acute kidney injury; DNI, delta-neutrophil index; NLR, neutrophil-to-lymphocyte ratio; MLR, monocyte-to-lymphocyte ratio; SMR, segmented-neutrophil-to-monocyte ratio; PLR, platelet-to-lymphocyte ratio; NLPR, neutrophil-to-lymphocyte-platelet ratio; SII index, systemic immune-inflammation index. Bold numbers indicate statistical significance. ^a^ Correction made for: age, gender. ^b^ Correction made for: age, gender, comorbidity score, baseline renal function, immune status. ^c^ Correction made for: age, gender, comorbidity score, baseline renal function, immune status, medication use. ^d^ Correction made for: age, gender, comorbidity score, baseline renal function, immune status, medication use, disease severity, provisional diagnosis in the emergency department.

**Table 4 jcm-11-01017-t004:** HR’s (95% CIs) for 30-day all-cause mortality.

	Univariate	Multivariate			
Ratios	Crude HR (95% CI)	Adjusted HR: Model 1 ^a^ (95% CI)	Adjusted HR: Model 2 ^b^ (95% CI)	Adjusted HR: Model 3 ^c^ (95% CI)	Adjusted HR: Model 4 ^d^ (95% CI)
**Modified DNI**					
Tertile 1, ≤0.5931	1.0 (reference)	1.0 (reference)	1.0 (reference)	1.0 (reference)	1.0 (reference)
Tertile 2, 0.5931–2.7088	0.8 (0.5–1.1)	0.9 (0.5–1.4)	0.8 (0.5–1.4)	0.8 (0.5–1.3)	0.8 (0.5–1.4)
Tertile 3, >2.7088	1.2 (0.8–1.9)	1.3 (0.8–2.0)	1.3 (0.8–2.1)	1.3 (0.8–2.1)	1.3 (0.8–2.0)
**NLR**					
Tertile 1, ≤4.2805	1.0 (reference)	1.0 (reference)	1.0 (reference)	1.0 (reference)	1.0 (reference)
Tertile 2, 4.2805–10.2276	1.3 (0.7–2.1)	1.1 (0.6–1.9)	1.2 (0.7–2.0)	1.2 (0.7–2.1)	1.2 (0.7–2.1)
Tertile 3, >10.2276	**1.9 (1.2–3.1)**	**1.6 (1.0–2.6)**	**1.7 (1.0–2.8)**	**1.8 (1.1–23.0)**	**1.7 (1.0–2.9)**
**MLR**					
Tertile 1, ≤0.5057	1.0 (reference)	1.0 (reference)	1.0 (reference)	1.0 (reference)	1.0 (reference)
Tertile 2, 0.5057–0.9830	0.9 (0.5–1.5)	0.8 (0.5–1.3)	0.8 (0.5–1.3)	0.8 (0.5–1.4)	0.8 (0.5–1.3)
Tertile 3, >0.9830	1.5 (0.9–2.2)	1.2 (0.7–1.9)	1.2 (0.7–1.9)	1.2 (0.8–2.0)	1.2 (0.7–1.9)
**SMR**					
Tertile 1, ≤6.7500	1.0 (reference)	1.0 (reference)	1.0 (reference)	1.0 (reference)	1.0 (reference)
Tertile 2, 6.7500–11.9633	1.6 (1.0–2.7)	1.5 (0.9–2.5)	1.5 (0.9–2.5)	1.5 (0.9–2.5)	1.5 (0.9–2.5)
Tertile 3, >11.9633	**1.9 (1.1–3.1)**	**1.8 (1.1–2.9)**	**1.9 (1.1–3.2)**	**2.0(1.1–3.3)**	**1.8 (1.1–3.1)**
**PLR**					
Tertile 1, ≤161.6468	1.0 (reference)	1.0 (reference)	1.0 (reference)	1.0 (reference)	1.0 (reference)
Tertile 2, 161.6468–314.6356	1.1 (0.7–1.9)	1.1 (0.6–1.8)	1.1 (0.6–1.8)	1.1 (0.6–1.9)	1.2 (0.7–2.1)
Tertile 3, 314.6356	**2.0 (1.2–3.2)**	**1.8 (1.1–2.9)**	**1.7 (1.0–2.8)**	**1.8 (1.1–2.9)**	**1.7 (1.1–2.8)**
**NLPR**					
Tertile 1, ≤1.9151	1.0 (reference)	1.0 (reference)	1.0 (reference)	1.0 (reference)	1.0 (reference)
Tertile 2, 1.9151–5.0605	**2.6 (1.5–4.4)**	**2.3 (1.3–3.9)**	**2.4 (1.4–4.2)**	**2.6 (1.5–4.5)**	**2.5 (1.4–4.4)**
Tertile 3, >5.0605	**2.2 (1.3–3.9)**	**1.8 (1.0–3.2)**	**2.0 (1.2–3.6)**	**2.2 (1.3–4.0)**	**2.1 (1.2–3.7)**
**SII index**					
Tertile 1, ≤869.43	1.0 (reference)	1.0 (reference)	1.0 (reference)	1.0 (reference)	1.0 (reference)
Tertile 2, 869.43–2414.46	0.6 (0.3–1.0)	0.5 (0.3–0.9)	0.5 (0.3–1.0)	0.6 (0.3–1.0)	0.6 (0.3–1.1)
Tertile 3, >2414.46	**1.9 (1.2–2.9)**	**1.7 (1.1–2.6)**	**1.7 (1.1–2.7)**	**1.8 (1.2–2.9)**	**1.8 (1.1–2.8)**

Abbreviations: HR, hazard ratio; AKI, acute kidney injury; DNI, delta-neutrophil index; NLR, neutrophil-to-lymphocyte ratio; MLR, monocyte-to-lymphocyte ratio; SMR, segmented-neutrophil-to-monocyte ratio; PLR, platelet-to-lymphocyte ratio; NLPR, neutrophil-to-lymphocyte-platelet ratio; SII index, systemic immune-inflammation index. Bold numbers indicate statistical significance. ^a^ Correction made for: age, gender. ^b^ Correction made for: age, gender, comorbidity score, baseline renal function, immune status. ^c^ Correction made for: age, gender, comorbidity score, baseline renal function, immune status, medication use. ^d^ Correction made for: age, gender, comorbidity score, baseline renal function, immune status, medication use, disease severity, provisional diagnosis in the emergency department.

## Data Availability

The data presented in this study are available on request from the corresponding author. The data are not publicly available due to privacy considerations.

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
