# Peer review of "Hematological Ratios Are Associated with Acute Kidney Injury and Mortality in Patients That Present with Suspected Infection at the Emergency Department"

_jcm, 2022, doi:10.3390/jcm11041017_

Round 1

Reviewer 1 Report

Thanks for allowing me to review an article which is well written and addresses an easily available marker to prognosticate the AKI outcome in the settings of infections.

However, I note certain isssues in the article that may warrant attention:

Overall comments:

SPACE cohort data is noted with PRIMARY outcome measure of AKI incidence based on values of S Creatinine. Limitations of S Creatinine measurements in sepsis should make a mention. Urine output was not measured or noted, despite being a factor of KDIGO defintion of AKI. I note, AKI present at ED were excluded, thus presence of pre-existing renal disease in patients who developed AKI is reflected in Tabel 1. 

 In animals, it is shown that Sepsis reduces muscle perfusion and thus the production of creatine falls, which blunts the increase in serum creatinine concentration and limits early detection of AKI. Together with dilutional effects secondary to aggressive fluid resuscitation in septic shock, AKI may be under-diagnosed. Similarly baseline S creatinine is a wide measure. 

Secondary outcome measure of mortality is quite straight forward, however cause of death would be helpful parameter, to associate the AKI, need for RRT and death in sepsis and its corelation with haematologic ratios. This association is less than clear in thie paper and could be explored. 

Haematological rations were calculated based on the single blood sample  collected at ED presentation. However, changes to these variables along the course of disease may vary. ASSOCIATION is more appropriate terminology than "RISK FACTOR" as in the title of the article.  This is used in mixed manner in ABSTRACT and in the substance of the article. Thus risk factor terminology needs to be changed. 

Advances in understanding the pathophysiologic mechanisms have provided insight into potential new therapies; however, effective, specific interventions for prevention and treatment of S-AKI (sepsis induced AKI) are still lacking. S-AKI is the result of a dysregulated response of the host to infection. Similarly, dusregulated response of the bone marrow to the infection is reflection within the Haematological ratio's. 

Thus, I note the death of 102  patients (160  AKI patients) within <30 days of presentation; is reflective of a very different cohort  with death from other causes than AKI  in the setting of suspected infection. Of note, 821 (43.3%) were immunocompromised, 98 (5.2%) had qSOFA score >2. 

The role on immunosuppression response to the bone marrow /white cells is well known, and considering the significant proportion of the cohort being immunosuppressed be clarified.  Leucopenia and blunted lymphocytic/ granulopoetic response from immunosuppression be further explored.  This is mentioned in the discussion, however should be mentioned earlier in the article. 

Cohort who developed AKI were more likely to have poor baseline eGFR, have hypertension, diabetes or CCF and on ACEi/ARB/ diuretics. This resonates with the general population, however almost all of Stage 1 AKI improve/ recover their renal function with time.  Thus mortality in this cohort is disproprotionately high for AKI and thus need for seperation of AKI and Mortality remains prudent. 

qSOFA score (Line 91) is an error - systolic BP (should be less than or equal to - not >).

HRs (CI) for AKI and Mortality:

Although it has reached statistical signifance in some of the rations; there is signifcant overlap in the CI in various tertiles to use these rations as "risk factor". These parameters should be considered as risk factor or an diagnostic indicatior (line 252) , if tested prospectively as a risk factors with sensitivity and specificty for each markers and in different AKI settings. 

Haematological parameters were not used to diagnose AKI or mortality in this study and has not affected the treatment parameters other than usual standrad of care. Thus its "potential use" could be explored. 

Reviewer 2 Report

Congratulations for your study

*Think you should improve the discussion part, it’s too long 

and there are repeated ideas. 
*Might be you could suggest algorithm for the Emergency room

teams to implement your results in identifying those high risk patients

Thank you  
